# “No Risk No Fun?”: Determinants of Satisfaction with Life in People Who Engage in Extreme and High-Risk Sports

**DOI:** 10.3390/ijerph192013328

**Published:** 2022-10-15

**Authors:** Daniel Krokosz, Mariusz Lipowski

**Affiliations:** Departament of Psychology, Faculty of Physical Culture, Gdańsk University of Physical Education and Sport, 80-336 Gdańsk, Poland

**Keywords:** psychology of sport, wellbeing, motivation

## Abstract

(1) Background: In this paper, we attempt to identify which personality and motivational variables significantly influence the sense of quality of life of individuals who practice extreme sports (ES) and high-risk sports (HRS). In addition, we examined to what extent these relationships are moderated by the athletes’ gender and experience in a given discipline. (2) Methods: A total of 363 individuals who practice ES or who practice HRS took part in the study. All of the participants were from Poland. Standardized questionnaires were used to measure the investigated personality traits (extraversion, neuroticism, psychoticism), motives, and satisfaction with life, characteristic of the practiced sports. (3) Results: A positive relationship was observed between levels of extroversion and sense of satisfaction with life in the groups of women who engage in HRS (*p =* 0.045) and men who engage in ES (*p* = 0.002). The tendency towards addiction was significantly negatively associated with sense of satisfaction with life only in men who engage in ES (*p* = 0.015). Individuals who engage in ES and HRS did not differ in levels of tendency towards addiction. (4) Conclusions: Individuals who practice ES differ from individuals who practice HRS in terms of personality features, motivations, and the determinants of their sense of satisfaction with life.

## 1. Introduction

The number of people enjoying alternative, often dangerous, sports has been growing in recent years [1,2]. Traditionally, such sport disciplines are referred to as high-risk sports, extreme sports, or adventure sports. They are defined as disciplines that are characterized by an inherent risk of injury (or even death), extremely intense physical effort, and the adventurous character of the undertaken activities [3]. The will to overcome difficulties, experience intense sensations, and control extreme situations is also emphasized [2].

Despite the fact that sports of this kind fascinate recipients and are the subject of scientific research, the attempt to capture the delicate relationships between predispositions, motivation, and life satisfaction still does not provide explicit scientific explanations. The presented paper aims to connect those areas of research.

The first point that needs to be clarified is the terminology used to describe this type of activity. The terms ‘high-risk sports’ and ‘extreme sports’ are often used interchangeably to refer to the same disciplines. However, in some cases, they are distinguished on the basis of differences in levels of associated danger. This can be observed mainly in the classifications made by insurance companies, which, using injury statistics, assess the dangers associated with given sports disciplines. According to the definition given by the Ergo Hestia travel insurance company, “extreme sports shall be understood as the participation in the expeditions or excursions to the places characterized by extreme climate or natural conditions […], as well as practicing the sports which require extraordinary skills, courage and operating in high-risk circumstances, often life-threatening […]” [4] (ERGO-Hestia; accessed on 30 June 2022). Cohen, Baluch, and Duffy [2] emphasize the importance of the empirical assessment of differences between those sports that are defined as extreme and those defined as high risk. Although insurance companies do not define HRS in precise way (they use only a list of disciplines (ERV/Ergo Hestia; Allianzdirect.pl)), HRS are understood as the disciplines that are more dangerous than common sports, less dangerous then ES, and are practiced in urban or non-extreme environments. The types of disciplines are often considered life-style sports (e.g., skateboarding, parkour, or windsurfing), while sports such as mountaineering, combat sports (Muay Thai or MMA), and motor sports would be considered ES. Since there is no clear distinguishing between those two categories among insurance companies, one of the main goals of this study is to determine whether those types of sports should be treated separately. This determination could contribute to better understanding of complexity of the phenomenon of ES and HRS and could also reveal the diversity of the personality determinants in people participating in these sports.

Quantitative exploration of the psychological profile of people who undertake extreme sports has usually examined the temperamental determinants for undertaking this kind of activity. Already by the 1970s, Zuckerman’s theory of sensation seeking played the dominant role in explaining the involvement in ES [5]. It suggests that people who engage in HRS and ES are characterized by higher levels of this feature than individuals who engage in other, safer sports disciplines and those who do not engage in sports at all [6]. However, more recent research suggests that individuals who engage in HRS and ES are not a homogenous group in terms of their temperament [7,8,9]. Kerr and Mackenzie [10] suggest that these discrepancies could be explained by the individual character of different HRS or ES—some activities last for only a few seconds (e.g., bungee jumping), while others may last for months (e.g., mountaineering or sailing solo around the world). Thus, sports in which the performance lasts a few seconds can be considered a source of strong stimulation, while sports in which the action takes much longer are more often associated with the ability to endure discomfort and sometimes even avoid stimulation [11]. Moreover, researchers stress the need to describe the personalities of extreme athletes in terms of a theory and tools different to the sensation seeking scale. Guszkowska and Bołdak [12], as well as Llewellyn, Sanchez, Asghar, and Jones [13], point out that the aforementioned example statement about distinguishing different groups of extreme athletes may be tautological—it is obvious that each individual who engages in such an activity will score high on a scale of seeking adventure and dangerousness, and differences between populations engaging in particular disciplines may not be observed due to a ceiling effect. The above premises, such as basing the diagnosis of the temperament of HRS and ES people on the SSS concept [5] and the current treatment of HRS and ES together, are the rationale for undertaking the presented research.

It can be assumed that, as in the case of personality, the introduction of a distinction between HRS and ES may reveal significant differences in the type of motivation of athletes. Since Mallory’s legendary “because it’s there” response, the fascination with why people engage in such dangerous activities has become a focus of both the media and researchers. The research in this area relies mostly on hermeneutic qualitative studies [14,15,16,17]. Conclusions from these studies indicate that the main motives are associated with overcoming challenges and difficulties, a sense of freedom, development of courage and humility, developing a special bond with nature, getting to know and learning to control one’s own emotions, and developing a sense of agency. These conclusions are in contrast to the stereotype often perpetuated by the media that the main (and maybe even only) motivation for undertaking extreme sports (ES) is sensation seeking [18]. Qualitative studies by Ewert, Gilbertson, Luo, and Voight [19] indicate that motivations for taking part in ES can vary depending on the gender of the athlete, the discipline in question, and the athlete’s level of experience. These motivations include social behaviors (being in a group, meeting people), seeking sensations, and developing one’s self-image. Frühauf et al. [14] indicate that the above motivations can definitely be considered to be psychological benefits of the sport one does, which may translate into a sense of wellbeing in individuals who engage in ES. However, research to date does not indicate how participation in HRS or ES is associated with a sense of satisfaction with life, especially taking into account gender differences. According to Chang [20], strong engagement in extreme sports is associated with the experience of the state of flow [21], especially for men. Despite the large number of studies describing the motivation of HRS and ES athletes, still few studies focus on the quantitative measurement of motivation and the relationship between motivation and personality traits. The presented study is based on the theory of motivational function of an objective by Zaleski and Lipowski [22,23]. According to this theory, the individual consciously and proactively chooses goals and plans their implementation in order to achieve well-being [22,23]. The key concept in the theory of Zaleski and Lipowski is the concept of goal (objective) in motivation. Zaleski defines objective as “a future state that is cognitively represented, possible to achieve, has a value and regulatory power, and is pursued by an individual through their actions” [23]. According to Zaleski’s theory, goals, when they are important for the individual, realistic to achieve, and are consistent with values, constitute a high motivational value and become visible in action.

However, it ought to be stressed that not all studies indicate beneficial consequences of extreme sports. Heirene, Shearer, Roderique-Davies, and Mellalieu [24] point out that extreme athletes, apart from the risk of serious injury, are also at risk of particular types of behavioral addiction [25]. Any time away from sports and obtaining the desired dose of sensation may lead to a withdrawal syndrome, which manifests as irritability and inability to experience pleasure. According to Willig [26], this is associated with a sense of stress and negative affect that may be compared to withdrawal symptoms in people addicted to substances.

The presented study aims to explore two fundamental questions about HRS and ES: who practices these sports and what constitutes the main motivations. Although these topics were often studied from a psychological perspective, they often focused on a single method (Zuckerman’s SSS) or explored motivations in a qualitative way. Due to these limitations of the previous studies, we decided to use a questionnaire that examines the biological basis of personality, which in an accurate way (avoiding the ceiling effect) can show the difference between HRS and ES. In addition, a questionnaire was used that quantitatively measures the validity of specific motives for participation in the disciplines in question. What is also important is that there is a lack of research on how the personality profile of HRS and ES practitioners, as well as the motives they choose, translate into their sense of satisfaction with life. In addition, an important reason for undertaking the presented research is to check whether fashion as a motive for undertaking HRS in particular may be related to the fact that people with a lower demand for stimulation can engage in this type of sport.

## Present Study

In the present project, we sought to identify which personality and motivation variables are important for sense of satisfaction with life in individuals who engage in extreme sports and high-risk sports. Additionally, we investigated to what extent these relationships are moderated by the participants’ genders and their experience in a given sport discipline. Two hypotheses were made: H1—the personality profiles and the motivations of extreme athletes differ from the personality profiles and the motivations of individuals who engage in high-risk sports; H2—personality profiles and motivations have different impacts on satisfaction with life in extreme athletes and high-risk sports athletes.

## 2. Materials and Methods

### 2.1. Participants

Participants consisted of individuals who have engaged in high-risk or extreme sports regularly for at least two years (inclusion into HR or ES groups was based on insurance companies’ classifications of disciplines as ES or HRS). Selection was purposeful, and the selection criteria were the abovementioned conditions: at least two years of experience of participation in HRS or ES and regularity of practicing a given discipline. The research was carried out in clubs, associations, and informal groups connecting practitioners of the type of sports discussed. The list of clubs and groups was created on the basis of HRS and ES organizations available in the Pomeranian Voivodeship in Poland. The limitation to Pomeranian Voivodeship was caused by the financial resources allocated to the study, which, due to the fact analysis of sensitive data, was carried out personally by a psychologist in the subject (it was not carried out in an online form). Then, by e-mail or telephone, visits from the person conducting the examination were arranged. A total of 25 clubs and groups were contacted, of which 19 clubs agreed to participate in the study. In addition, in order to reach the largest number of respondents, the snowball method was used, which allowed for the recruitment of additional people. Given the difficulty of determining the size of the entire population of HRS and ES practitioners in the study area, it was difficult to accept a minimal research sample. Therefore, it was decided to reach the largest possible group of HRS and ES enthusiasts. A total of *N* = 363 individuals took part in the study, including *n* = 120 women and *n* = 243 men. The mean age of the participants was *M* = 24.20 years (*min* = 16; *max* = 52; *SD* = 5.33). On average, participants had been practicing their discipline for *M* = 5.33 years (*min* = 2 years; *max* = 29 years; *SD* = 3.56), devoting *M* = 25.67 h to it per month (*min* = 4 h; *max* = 100; *SD* = 15.09). In case of seasonal sports, only athletes who practiced their sport for two months per year were taken into account. Among the participants, *n* = 184 (50.69%) individuals practiced sports categorized as extreme sports (ES), and *n* = 179 (49.31%) individuals engaged in sports categorized as high risk (HRS). The categorization was made on the basis of an insurance companies list (ERV, Ergo Hestia, Allianzdirect.pl) and the assessment of competent judges (*N* = 5), specifically, experienced, retired extreme sportsmen who helped to decide about the categorization in conflict cases (e.g., snowboarding was considered HRS because participants were not practicing the freeride form of this sport). The following disciplines were categorized as high risk on the basis of insurance companies’ classifications: parkour, BMX, skateboarding, snowboarding, windsurfing, kitesurfing, and boxing. The following disciplines were classified as extreme sports: MMA, Muay Thai, Krav Maga, kickboxing, mountaineering, freeride skiing/biking, motocross, off-road, parachuting, and paragliding. The following categories of disciplines were practiced by the participants: city (skateboarding; roller skating, parkour), *n* = 80; motor (off-road, motocross), *n* = 18; aerial (parachuting, paragliding), *n* = 32; cycling (freeride, downhill), *n* = 40; martial arts (MMA, Muay Thai, Krav Maga, Brazilian jiu-jitsu, kickboxing), *n* = 76; water (wakeboarding, kitesurfing, skimboarding, mountain kayaking), *n* = 56; winter (freeride skiing, snowboarding), *n* = 32; and mountain (Himalayan mountaineering, alpinism, mountaineering) sports *n* = 29.

All subjects signed a consent form in order to participate in the study. In the case of participants under 18 years old, additional consent from their parents was obtained. The study was reviewed by The Ethics Board for Research Projects at the Institute of Psychology, University of Gdańsk, no. 8/2014.

### 2.2. Instruments

The Polish adaptation [27] of Eysenck’s Personality Questionnaire (EPQ-R) was used [28] to assess innate predispositions for taking part in extreme sports. The EPQ-R questionnaire was used because it refers to biological determinants of temperament. Furthermore, it was important for the measurement tools used in this study to be sensitive to the entire spectrum of the variability of a variable, in order to reveal potential differences within a group. Taking this into account, we chose not to use Zuckerman’s Sensation Seeking Scale (SSS) [29], which, as has already been described in the theoretical part of this paper, often yields extremely high results in the case of extreme athletes. The EPQ-R measures (1) Extraversion; (2) Neuroticism; (3) Psychoticism; (4) Lying; (5) Addiction; and (6) Criminality. Participants responded to 106 questions (e.g., Do you like meeting new people? Do you suffer from insomnia?) by providing an answer of yes or no, depending on whether the sentence referred to their personality. The raw scores were converted to sten norms, which were interpreted as follows: 1–3 low score; 4–7 mean score; 8–10 high score [30]. All of the listed scales had Cronbach’s alphas above 0.75.

The Inventory of Physical Activity Objectives (IPAO) [22,31] was used in order to determine motives for involvement in extreme and high-risk sports. The respondent answers questions regarding their involvement in competitive sports (both present and previous) as well as the forms and intensity of their physical activity. The IPAO includes 12 objectives that are accompanied by a Likert scale (1–5), and the respondent is asked to assess the importance of the listed objectives, where 1 stands for completely unimportant and 5 for very important. After consulting competent judges and the authors of the original questionnaire, a list of 12 motivations was adjusted from original instrument to better suit the specifics of HRS and ES. Preliminary pilot studies showed that subjects often added missing, specific motives for HRS and ES, while rarely evaluating goals no. 10, 11, and 12 as important. On the other hand, with goal 6—pleasure from physical activity, they often added the comment that the feelings accompanying HRS and ES, although associated with pleasure, are more associated with thrill and adrenaline. After consultation with the competent judges and authors of the original questionnaire, it was decided that we would replace questions 6, 11, and 12 with the more adequate ones proposed by the experts and the pilot group. We decided to leave question 10 unchanged due to its possible adequacy to fashionable HRS. Their importance was assessed by the subjects: (1) a fit, shapely body (beauty, sculpted and firm body); (2) physical fitness, being ‘in shape’; (3) the company of other people; (4) managing stress; (5) escape from everyday life; (6) extreme experiences, adrenaline (changed from ‘pleasure from physical activity’); (7) boosting one’s confidence, overcoming one’s weaknesses; (8) health and wellbeing; (9) fashion; (10) gaining the appreciation of peers, wanting to prove oneself; (11) being together with nature (changed from ‘fulfilling the need for activity’); and (12) finding the limits of human ability (changed from ‘promoting physical activity’). Additionally, if the goal which was the most important for a given participant was not on the list, there was the option to add it. After a participant identifies their most important goal, they then assess 18 statements selecting the most applicable answers on a scale from 1 to 5. In addition to measuring the attitude of the subjects towards particular goals, the individual scores on the Likert scales are summed up. The final score indicates the importance of the heterogeneity of the objectives one sets. The Cronbach’s α reliability coefficient for this version was 0.78. The IPAO questionnaire was used because it allows for the precise assessment of the importance of selected motivations. Importantly for the current study, it allows for the quantitative assessment of an athlete’s motivations for engaging in HRS and ES.

The Satisfaction with Life Scale (SWLS) scale by Diener, Emmons, Larson, and Griffin [32], with a Polish adaptation by Juczyński [33], was used to measure participants’ satisfaction with life. The subject noted how much they agree with the five statements about life satisfaction (e.g., I’m satisfied with my life) on the Likert scale from 1 (I completely disagree) to 7 (I completely agree). Overall satisfaction with life is calculated as the sum of all scores, which can be compared with sten norms. The Cronbach’s alpha reliability index for this tool was 0.85, and satisfactory validity was confirmed by correlating the SWLS scores with scores on other scales measuring satisfaction with life. This tool is commonly used and, importantly for the participants, filling in the questionnaire takes very little time.

### 2.3. Data analysis

Statistical analyses were conducted with the Polish version of STATISTICA 12 (TIBCO, Palo Alto, CA, USA) and included general linear models, multiple regression, and indicator variables in regression. General linear models 2 × 2 ANOVA was used to examine the differences in personality profiles and motives of practicing sport with respect to gender (male or female) and sport discipline (HRS or ES). Multiple regression was used to determine whether personality traits and motivations influenced the sense of satisfaction in life of the athletes. Indicator variables in regression were used to examine the differences between genders and sport disciplines in the relationships of personality traits and motivation with life’s satisfaction. The threshold for the level of significance in statistical calculations was 0.05.

## 3. Results

### 3.1. Personality Traits

The first step of analysis was to assess the personality features of participants engaging in ES and HRS. In order to compare the personality traits of men and women who engage in various disciplines, a 2 × 2 ANOVA (gender × sport discipline) was conducted. There was an effect of gender on the Psychoticism scale, *F*(4, 357) = 8.77; *p* < 0.001. On this scale, men (*M* = 12.67; *SD* = 5.78) scored higher (post hoc: *p* = 0.024) than women (*M* = 11.10; *SD* = 5.81). The opposite relationship was observed for Neuroticism, where women (*M* = 12.80; *SD* = 5.40) scored significantly higher than men (*M* = 10.42; *SD* = 5.85); post hoc: *p* < 0.001. An effect of discipline was observed on the Psychoticism and Extraversion scales, *F*(4, 357) = 5.12; *p* = 0.001. Individuals who engage in high-risk sports scored higher on the Psychoticism scale than individuals who engage in extreme sports, post hoc: *p* < 0.001. In the case of Extraversion, there was the opposite relationship. Individuals who engage in extreme sports were characterized by higher Extraversion than those who engage in high-risk sports, post hoc: *p* < 0.030. Moreover, women also scored higher than men on the Addiction scale (post hoc: *p* = 0.012). The effect of interaction between gender and sport on personality traits was not significant.

### 3.2. Motivation

Analysis of the frequencies of motivations revealed that physical fitness was the most important goal for both men and women. This goal was the most important for 22.06% of women and 32.48% of men. The second most frequently selected goal was having a fit, shapely body (beauty, sculpted and firm body)—16.18% for women and 15.29% for men. For women, the third most important goal was health and wellbeing (14.71%), while for men it was escape from everyday life (8.92%).

In order to analyze the differences in the assessments of the importance of motivations for engaging in physical activity, a 2 × 2 factorial ANOVA was conducted (gender × sport discipline). The effect of gender turned out to be significant in the case of three motivations, *F*(17, 326) = 2.34, *p* = 0.002.

The first was having a fit, shapely body (beauty, sculpted and firm body), which was more important for women (*M* = 4.29; *SD* = 0.76) than men (*M* = 3.81; *SD* = 1.11; post hoc: *p* < 0.001). Likewise, boosting one’s confidence and overcoming one’s weaknesses was also more important for women (*M* = 4.22; *SD* = 0.95) than for men (*M* = 3.87; *SD* = 1.06; post hoc: *p* < 0.001). The last of the differences in motivations was observed in the case of health and wellbeing. As in the case of the first two, it was more important for women (*M* = 4.48; *SD* = 0.70) than for men (*M* = 4.24; *SD* = 0.89; post hoc: *p* = 0.027).

An effect of the type of sports was revealed in the case of assessments of the importance of five motives: physical fitness and being in shape; boosting one’s confidence, overcoming one’s weaknesses; health and wellbeing; fashion; and finding the limits of human ability, *F*(17, 339) = 2.32; *p* = 0.002. In comparison with individuals who engage in HRS (*M* = 4.30; *SD* = 0.87), ES athletes (*M* = 4.57; *SD* = 0.71) placed the physical fitness and being in shape goal higher (post hoc: *p* = 0.001). The goal of boosting one’s confidence and overcoming one’s weaknesses was also assessed as more important by ES athletes (*M* = 4.10; *SD* = 0.98) than by individuals who engage in HRS (*M* = 3.87; *SD* = 1.08; post hoc: *p* = 0.034). A similar relationship was observed for the health and wellbeing goal. Individuals who engage in extreme sports (*M* = 4.45; *SD* = 0.66) assessed this goal as being more important than those who engage in HRS (*M* = 4.19; *SD* = 0.98; post hoc: *p* = 0.004). The last goal that was assessed as more important by ES athletes (*M* = 4.08; *SD* = 1.03) was finding the limits of human ability (post hoc: *p* = 0.020)—individuals engaging in HRS assessed the importance of this goal as *M* = 3.82; *SD* = 1.24. Only in the case of the fashion goal did individuals who engage in HRS (*M* = 2.76; *SD* = 1.36) score higher than those who engage in ES (*M* = 2.38; *SD* = 1.30; post hoc: *p* = 0.005). The effect of interaction between gender and sport on personality traits was not significant.

### 3.3. Satisfaction with Life

The next step of the analyses involved the assessment of the degree to which the participants were satisfied with life. Among women, those with high levels of satisfaction with life (35.83%) were significantly more numerous than those with low levels (10.00%; *p* < 0.001). A similar situation was observed in the case of men, where 24.36% of individuals had high levels of satisfaction with life and only 8.64% were dissatisfied (*p* < 0.001). In terms of gender differences, it turned out that the percentage of women who engage in ES and are highly satisfied with life was significantly higher than men (*p* = 0.023). There were, however, no differences in the percentage of individuals with low levels of satisfaction with life.

A 2 × 2 ANOVA (gender × type of sports) revealed no gender differences in average levels of satisfaction with life; however, the effect of type of sports was significant: *F*(1, 360) = 8.74; *p* = 0.003. Individuals engaging in less risky sport disciplines were characterized by higher levels of satisfaction with life (*M* = 23.98; *SD* = 5.50) than extreme athletes (*M* = 22.09; *SD* = 6.04); post hoc: *p* = 0.002.

### 3.4. Personality Traits and Satisfaction with Life

Multiple regression analysis was conducted to examine the relationship between personality features and sense of satisfaction with life in individuals who engage in ES and HRS. In the analysis, it turned out that after including variables such as age, lying, and criminality, the R2 index was higher than when these variables were not included. Only in the case of women who engage in HRS was no relationship observed between personality features and satisfaction with life. In all other cases, the model assuming personality determinants of satisfaction with life was significant. The regression model is summarized in Table 1, and descriptive statistics are shown in Table 2.

Next, dummy variable analysis was used to examine differences regarding the influence of particular personality features on sense of satisfaction with life in the groups, on the basis of gender. The analysis revealed that only age and experience variables had different influences on sense of satisfaction with life in the compared groups. Among women who engage in HRS, age determined sense of satisfaction with life to a significantly higher degree than in women who engage in ES (t = 2.32; *p* = 0.021). Differences in the influence of experience on sense of satisfaction with life were also observed—experience determined the sense of satisfaction with life to a higher degree in women who engage in ES than in men who engage in ES (t = 2.47; *p* = 0.014).

### 3.5. Motivation and Satisfaction with Life

Next, we analyzed the ways in which the goals formulated by the athletes translate into their sense of satisfaction with life. Interestingly, the model turned out to be significant in both men and women who engage in ES. Detailed results of the regression analysis are presented in Table 3, and descriptive statistics are provided in Table 4.

A dummy variable analysis based on gender revealed no group differences in the assessments of separate motivations.

## 4. Discussion

The results justify the distinction between high-risk sports and extreme sports. This classification, proposed by insurance companies, was reflected in both the subjective perceptions of individuals who take part in such activities, as well as in differences regarding their predispositions and motivations. The first hypothesis was confirmed mostly in the group of men who engage in ES. Surprisingly, extraversion equally strongly determined satisfaction with life in women who engage in HRS. This result can be explained with reference to a meta-analysis that indicates that men are characterized by a significantly higher need for stimulation and sensation seeking [34]. Thus, it can be speculated that ES provides the optimal level of risk for extroverts, while for women this level may be beyond the optimum, making this relationship insignificant. This explanation is also applicable to the relationships observed in the HRS group. Here, a significant influence of extraversion on sense of satisfaction with life was observed only for women (optimal levels of risk), while in men, there was no relationship between these variables (not enough risk). Of course, this explanation requires further examination in future studies.

Thanks to the use of the EPQ-R tool, we did not observe extremely high results on either the Extraversion scale, which measures need for stimulation, or on the Psychoticism scale, which is associated with impulsivity and acting on the border of social adjustment. Frequency analysis for high, low, and moderate results on each of the subscales allowed us to observe that some individuals who engage in HRS and ES are characterized by low levels of extraversion and psychoticism, which, to date, has not been accounted for in the literature. Moreover, when analyzing mean sten scores for each of the groups, we were able to see that the levels of extraversion were average, which confirms the conclusions of Guszkowska and Bołdak [12] as well as Barlow et al. [7], who suggest that individuals who engage in HRS and ES are non-homogeneous in terms of their temperament and that their predispositions should be measured by a tool other than Zuckerman’s SSS [29]. Moreover, these previous findings may suggest that, due to the increasing popularity and attractiveness of extreme sports [1], more individuals who are not particularly predisposed to such activities are undertaking them, which may indicate that it is not only personality that determines participation in these types of activities. Although exploring these determinants in a changing world should be the subject of future research, it can be speculated that people who are not personally assigned to ES or HRS may choose these types of activities because of possible social benefits such as popularity or social media outreach.

It is extremely important that among all the individuals who engage in ES and HRS, there were practically no individuals characterized by low predisposition towards addiction. Heirene, Shearer, Roderique-Davies, and Mellalieu [24], in a qualitative study conducted on a group of Himalayan mountaineers, found that individuals who practice such disciplines exhibit all of the symptoms of withdrawal syndrome. Barlow, Woodman, and Hardy’s [7] work suggests that the need to pursue stimulation among Himalayan mountaineers is lower in comparison with other sports. Thus, it might be that in the case of more stimulating sports, the symptoms of addiction are even stronger. Schüler, Wegner, and Knechtle [35], in a study on individuals who engage in extreme endurance sports, found a positive correlation between addiction and the intensity of training, as well as a negative correlation with age. Undoubtedly, this is another argument for the addictiveness of the intense emotions associated with HRS and ES. A negative influence of tendencies towards addiction on sense of satisfaction with life was confirmed, but, again, the relationship was observed only in men who engage in ES. This result is comparable with the observations of Chang [20] regarding the relationship between levels of engagement in ES and the particular ease with which men reach the state of flow, while this relationship was weaker in women. Experiencing positive emotional reinforcement in this state may have a stronger influence on the tendencies towards addiction. However, the assumption that individuals who engage in ES would score higher on the scale measuring predispositions toward addiction than individuals who do HRS was not confirmed. None of the listed motivations were related to the sense of satisfaction with life of individuals who do ES and HRS. However, again, in the group of women who engage in HRS and men who engage in ES, it was observed that in terms of motivations associated with extreme experiences, adrenaline has the strongest relationship with sense of satisfaction with life.

The result indicating that women score higher on the addiction scale than men may be surprising at first glance. However, if we assume that they experience emotions much more intensely than men [36], then, in effect, it may also be easier for them to become addicted—but this explanation undoubtedly requires further investigation. This lack of ambiguity in the obtained results can be explained by the limitation of the presented study—the tool used to measure the tendency towards addiction. The scale used is only an experimental tool, not one aimed at accurately measuring addiction in extreme athletes, such as the scale proposed by Ahn, Cho, and So [37].

When analyzing the motivations of individuals who engage in ES and HRS, one ought to first of all into account that in the extreme experiences, the adrenaline goal was not assessed by the participants as the most important. This result, at least at a declarative level, is in contrast to critical voices who argue that the main reason for taking part in ES and HRS is an irrational craving for intense emotions and a sense of increased arousal. For the participants of this study, the most important goals were associated with physical fitness, having a fit body, and health, and thus they were no different from those most commonly indicated by individuals who engage in forms of physical activity and sport that are not associated with increased risk [31]. This may be caused by the fact that individuals who engage in HRS and ES do not assess their sport disciplines as extremely dangerous [38], and thus their attitude is not significantly different from that of people who practice non-risky disciplines. One can speculate that providing oneself with desired levels of stimulation from risky forms of physical activity may thus be a latent and indirect motive, one that is not expressed in declarative statements about goals typical of any physical activity.

The gender differences revealed in the results of the assessments of motivations, especially those regarding the fit body and health goals, are in line with previous studies [39].

Again, the importance of distinguishing between ES and HRS was confirmed in the differences observed in the assessments of the importance of particular goals. It is notable that goals that are more important for ES were associated with a responsible and professional approach towards sport. This attitude is understood as taking care of proper preparation, warm-up, training, and diet. The higher assessment of goals related to physical fitness and health indicates that they are treated as resources that may be decisive in avoiding serious injuries in critical situations. Health and physical fitness seem to be being thought of as instruments—things without which it would be impossible to practice the sport responsibly. This is in line with the conclusions of the qualitative studies by Brymer and Schweitzer [16], who found that engaging in ES is an activity that changes an individual, increasing their humility and ability to combat fear, as well as teaching them to treat their life responsibly and respect their health. Higher scores for boosting one’s confidence and overcoming one’s weaknesses are not surprising—a person’s confrontation with the elements, or an intense fight with an opponent in the case of martial arts, is undoubtedly a challenge that can inspire self-development and improve one’s sense of self-efficacy. The motivation associated with finding the limits of human ability, which was also assessed higher by those who engage in ES, indicates that this group places more emphasis on transgression and goals that go beyond sensual experiences or intense emotions. Similar conclusions were drawn by Lebeau and Sides [40], who pointed out that ES are more often associated with motives connected to transgression than are mainstream sport.

The fashion motivation was assessed as relatively unimportant by individuals from both ES and HRS groups. However, one should note that this was the only goal whose importance was significantly higher for individuals engaging in HRS. This may suggest that this type of sport is more associated with an attractive image. Furthermore, sports such as skateboarding, BMX, or parkour [41] are often considered a subculture and may be a part of one’s identity. Engaging in niche sport disciplines definitely allows a person to be different from their peers and to identify with a group of people who have a passion or their own characteristic style. A particular paradox should be noted: youths who pursue originality through being a part of a subculture undergo a kind of uniformization and become targets for commercial concerns [42,43].

The fact that individuals who engage in HRS were characterized by higher levels of satisfaction with life than those who engage in ES could be explained by the specifics of the two categories of sport. This has been confirmed by the studies described in the doctoral thesis of Sidorova [44], who found that surfers (who, in line with the classifications of insurance companies, are representatives of HRS) score higher on most dimensions of the satisfaction with life scale than individuals who do not do HRS. Moreover, research has shown that surfers are characterized by much higher levels of autotelic activities than individuals who do not engage in such sport, which may be associated with global levels of satisfaction with life [45].

## 5. Conclusions

The main conclusions of the presented research indicate that the division into HRS and ES is adequate and effective. Both groups of athletes differ not only in personality traits and motives but also in the determinants of the feeling of satisfaction with life. Attempting to identify the psychological consequences of engaging in HRS and ES is important for the further development of this area of research. Examining changes in motivations by using longitudinal studies could make it possible to look at the phenomenon in a more accurate way. Motivation is a dynamic process that changes with athletes’ experience and is undoubtedly associated with critical events such as injuries or threats to one’s life. Including such variables in the model and making several measurements could be an interesting addition to knowledge regarding the specifics of motivations for taking part in HRS and ES. The presented study did not avoid limitations. Purposeful selection for the sample, as well as limitation to one region in Poland, may be an obstacle in extrapolating the obtained results to the entire population. Furthermore, lack of scientific definition of HRS and ES remains a topic to be resolved in subsequent studies. In addition, the measurement of sport addiction in subsequent studies should be developed using a more precise tool.

## Figures and Tables

**Table 1 ijerph-19-13328-t001:** Summary of the regression analysis: personality features and satisfaction with life.

	♀ HRS		♂ HRS		♀ ES		♂ ES	
	β	*p*	β	*p*	β	*p*	β	*p*
Psychoticism	0.30		0.15		0.18		0.12	
Extraversion	0.37	0.045	0.11		0.06		0.33	0.002
Neuroticism	–0.01		0.02		0.38		–0.26	
Lying	0.32		0.40	0.007	0.18		0.46	<0.001
Criminality	0.49		0.16		–0.29		0.69	0.020
Addiction	–0.40		–0.41		–0.01		–0.58	0.015
Age	0.32	0.014	0.11		–0.11		0.02	
Experience	0.14		0.09		0.50	0.002	0.10	
Summary of the regression analysis	*F* = 2.37 *R^2^* = 0.25 *p* = 0.028		*F* = 2.46 *R^2^* = 0.16 *p* = 0.018		*F* = 2.33 *R^2^* = 0.29 *p* = 0.035		*F* = 4.55 *R^2^* = 0.23 *p <* 0.001	

**Table 2 ijerph-19-13328-t002:** Descriptive statistics of personality traits.

	♀ HRS		♂ HRS		♀ ES		♂ ES	
	*M*	*SD*	*M*	*SD*	*M*	*SD*	*M*	*SD*
Psychoticism	12.68	6.69	13.88	5.50	9.17	3.75	11.62	5.83
Extraversion	14.70	5.50	13.58	5.22	15.50	4.96	15.02	4.99
Neuroticism	12.79	5.65	10.95	5.94	12.81	5.13	9.96	5.75
Lying	9.24	3.62	8.90	4.37	9.44	3.72	8.57	4.54
Criminality	6.42	1.54	6.47	1.62	5.78	1.20	6.13	1.64
Addiction	15.50	4.12	14.28	4.24	15.02	3.81	13.66	4.41

**Table 3 ijerph-19-13328-t003:** Summary of regression analysis: importance of goals and satisfaction with life.

	♀ HRS		♂ HRS		♀ ES		♂ ES	
	β	*p*	β	*p*	β	*p*	β	*p*
Fit body	0.04		–0.03		0.07		0.16	
Physical fitness	–0.10		–0.26	0.039	0.08		0.06	
Company of others	0.00		0.02		–0.47	0.003	0.03	
Managing stress	0.05		0.09		0.05		–0.23	
Escape	–0.32		–0.24		0.00		–0.03	
Extreme sensations	0.43	0.037	0.11		0.10		0.45	<0.001
Boosting confidence	0.05		–0.05		–0.10		–0.25	0.016
Health	0.09		0.06		–0.19		0.15	
Fashion	–0.16		0.04		0.20		0.07	
Appreciation from peers	0.12		–0.18		–0.24		0.05	
Commune with nature	0.14		0.00		0.15		0.09	
Finding limits	–0.10		0.01		0.10		0.05	
Age	0.43	0.022	0.20		–0.16		0.09	
Experience	0.10		0.06		0.60	<0.001	0.12	
Summary of the regression analysis	*F* = 1.39		*F* = 1.37		*F* = 2.58		*F* = 3.57	
	*R^2^* = 0.28		*R^2^* = 0.17		*R^2^* = 0.49		*R^2^* = 0.31	
	*p* = 0.193		*p* = 0.182		*p* = 0.010		*p <* 0.001	

**Table 4 ijerph-19-13328-t004:** Summary of regression analysis: importance of goals and satisfaction with life.

	♀ HRS		♂ HRS		♀ ES		♂ ES	
	*M*	*SD*	*M*	*SD*	*M*	*SD*	*M*	*SD*
Fit body	4.33	0.75	3.66	1.12	4.24	0.78	3.94	1.09
Physical fitness	4.41	0.82	4.23	0.89	4.68	0.64	4.53	0.73
Company of others	3.95	0.97	3.95	0.96	4.04	0.95	3.81	1.11
Managing stress	4.26	0.95	4.17	0.96	4.30	0.96	4.22	0.98
Escape	3.92	1.15	4.12	1.05	4.07	1.08	4.05	1.07
Extreme sensations	4.21	1.00	4.11	1.10	4.31	0.84	4.23	1.11
Boosting confidence	4.08	1.08	3.74	1.08	4.39	0.76	3.98	1.04
Health	4.41	0.70	4.06	1.09	4.56	0.69	4.40	0.64
Fashion	2.64	1.20	2.83	1.45	2.72	1.34	2.23	1.25
Appreciation from peers	2.91	1.22	2.72	1.30	3.22	1.25	2.95	1.25
Commune with nature	3.71	1.08	3.32	1.17	3.46	1.36	3.71	1.27
Finding limits	3.68	1.30	3.90	1.21	4.15	0.96	4.05	1.07

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
