# Peer review of "“No Risk No Fun?”: Determinants of Satisfaction with Life in People Who Engage in Extreme and High-Risk Sports"

_ijerph, 2022, doi:10.3390/ijerph192013328_

Round 1

Reviewer 1 Report

** Page 1, Line 30: "controllng" --> "control" 

** Page 1, Line 30: "are" --> "is"

** Page 1, Line 32: "disciplines, however" --> "disciplines. However," or "disciplines. In some cases, however,"

** Page 1, Line 34-35: Please check grammatical correctness of the sentence

** Page 1, Line 42:  Please check grammatical correctness of the sentence

** Page 2, Line 1~2: Please check grammatical correctness of the sentence

** Page3, Line 100~101: "motivations" --> "the (level, kind, or what) of motivation" "individuals" --> "the individuals" 

** Please correct grammatical error in a number of sentences from Abstract to the end. 

** METHOD: The validity and the reliability of all the questionnaires adopted in this study need to be examined before distribution.    

Author Response

Response to the reviewer

Thank you for pointing out the errors which were corrected in our paper. Furthermore, the text was checked by a native speaker to correct grammatical errors.

All tests adopted in the study were examined before distribution. Cronbach Alpha is provided for every test.

** Page 1, Line 30: "controllng" --> "control" 

** Page 1, Line 30: "are" --> "is"

** Page 1, Line 32: "disciplines, however" --> "disciplines. However," or "disciplines. In some cases, however,"

** Page 1, Line 34-35: Please check grammatical correctness of the sentence

** Page 1, Line 42:  Please check grammatical correctness of the sentence

** Page 2, Line 1~2: Please check grammatical correctness of the sentence

** Page3, Line 100~101: "motivations" --> "the (level, kind, or what) of motivation" "individuals" --> "the individuals" 

** Please correct grammatical error in a number of sentences from Abstract to the end. 

** METHOD: The validity and the reliability of all the questionnaires adopted in this study need to be examined before distribution.    

Reviewer 2 Report

In this study, the authors explored the psychological characteristics of extreme and high risk sport athletes. I believe the manuscript is generally well written and the research procedures appear sound. That being said, I believe there are meaningful concerns that need to be addressed before the manuscript can be considered for publication. Most importantly, a stronger rationale for the current research needs to be developed. I will highlight my concerns in more detail below.

I am also wondering how the current study aligns with the Aims and Scope of the International Journal of Environmental Research and Public Health. Specially. I am not sure about the link with (public) health.

ABSTRACT

Specific comments:

l. 11: Remove comma. Also, please change to “who practice ES and HRS” (should this be and/or?).

ll. 13-14: I recommend listing the variables instead of the instruments.

ll. 15-19: Please list p-values.

INTRODUCTION

While the Introduction provides helpful background information for the current study, it currently fails to establish a strong rationale for the research. Specifically, what are the gaps/limitations in previous literature that are being addressed. In other words, why was the current study needed? The information in lines 92-94 is not sufficient in establishing this rationale.

Specific comments:

ll. 25-47: I think it would be helpful to list examples of each type of sport. Also, is there a more academic definition that could be used to conceptualize these sports? Lastly, the authors state that “there is no clear distinguishing between those two categories among insurance companies.” Is there one in the literature?

ll. 47-49: The authors state that “one of the main goals of this study is to determine whether those type of sport should be treated separately.” Please provide a rationale.

ll. 50-91: While the authors provide a nice overview of previous research here, I am left wondering what the current study is adding to the literature. In other words, what gaps/limitations are being addressed by this study? A stronger rationale is needed.

ll. 55-56: In what way?

ll. 96: Personality and motivation are extremely broad concepts that each include many more specific constructs. For example, what motivation theory are the authors basing the current study on? I think it is necessary to specify further in the Introduction (which would also allow to provide conceptual explanations).

METHOD

Specific comments:

l. 104: Please add a data analysis section.

ll. 109-110: Based on what criteria?

ll. 110-113: More detail is needed regarding the sampling and recruitment procedures (e.g., how were clubs selected, how were individuals contacted, what was the response rate?).

l. 115: Was parental consent obtained for participants who were younger than 18 years old?

l. 118: I suggest using a different term than “sportsmen” since the sample also included females.

ll. 140-180: Please make sure to provide response format, example items, and psychometrics for all instruments.

ll. 157-158: The author states that “a list of twelve motivations was created and their importance was assessed by the subjects” (ll. 157-158). How does this differ from the original instrument and why were modifications done?

RESULTS

Specific comments:

Tables 1 and 2: I think it is necessary to provide descriptive statistics for all assessed variables.

ll. 259-260 and 274-275: Based on gender?

DISCUSSION

The Discussion includes various interpretations that are not supported by empirical evidence. I would like to see the author provide such links when offering explanations for the current findings.

Specific comments:

ll. 280-307: So how do you interpret the differences between men and women in your sample? I do not believe that ll. 282-284 is sufficient in explaining the current findings.

ll. 295-301: Please clarify as these statements seem to contradict each other. Then, please also revise the rest of this paragraph (ll. 302-307) for clarity.

l. 308: What constitutes a “high result?”

ll. 308-399: Please split this extremely long paragraph.

l. 315: Surprising in what way?

l. 321: “state of affairs”?

ll. 321-323: Would this not suggest that personality does not determine participation then?

ll. 323-334: How do you interpret this relationship? Please go beyond the descriptive comparison to previous studies.

ll. 347-352: Please provide supporting evidence.

ll. 356-357: Please explain further and provide supporting evidence.

ll. 361-362: What constitutes a “responsible and professional approach towards sport?”

ll. 380-384: Please interpret further and provide supporting evidence.

ll. 348-385: Please link with current findings.

ll. 386-389: Please clarify.

ll. 391-392: Please provide reference.

Author Response

Response to reviewer 2

Thank you very much for in-depth and substantive review of our paper. We agree with every point you’ve mentioned and made changes according to the specific points of the review. Here are the detailed answers to the Reviewer.

ABSTRACT

Specific comments:

  1. 11: Remove comma. Also, please change to “who practice ES and HRS” (should this be and/or?).
  2. 13-14: I recommend listing the variables instead of the instruments.
  3. 15-19: Please list p-values.

We’ve made the changes according to the reviewers suggestion

INTRODUCTION

While the Introduction provides helpful background information for the current study, it currently fails to establish a strong rationale for the research. Specifically, what are the gaps/limitations in previous literature that are being addressed. In other words, why was the current study needed? The information in lines 92-94 is not sufficient in establishing this rationale.

We have added additional line with rationale of the study (lines 108-111)

Specific comments:

  1. 25-47: I think it would be helpful to list examples of each type of sport. Also, is there a more academic definition that could be used to conceptualize these sports? Lastly, the authors state that “there is no clear distinguishing between those two categories among insurance companies.” Is there one in the literature?

Unfortunately we didn’t find academic definition of these sports, we have added this as a limitation to presented study. (line 478)

  1. 47-49: The authors state that “one of the main goals of this study is to determine whether those type of sport should be treated separately.” Please provide a rationale.

Rationale was provided in lines 53-55

  1. 50-91: While the authors provide a nice overview of previous research here, I am left wondering what the current study is adding to the literature. In other words, what gaps/limitations are being addressed by this study? A stronger rationale is needed.

We have added additional line with rationale of the study (lines 108-111)

  1. 55-56: In what way?

 Possible explanation was added (66-69)

  1. ll. 96: Personality and motivation are extremely broad concepts that each include many more specific constructs. For example, what motivation theory are the authors basing the current study on? I think it is necessary to specify further in the Introduction (which would also allow to provide conceptual explanations).

We’ve introduced theory of Motivational function of an objective as a base of the theory of motivation (87-90)

METHOD

Specific comments:

  1. 104: Please add a data analysis section.

Section was added. 213-215

  1. 109-110: Based on what criteria?

Criteria was specified more clearly 128

  1. 110-113: More detail is needed regarding the sampling and recruitment procedures (e.g., how were clubs selected, how were individuals contacted, what was the response rate?).

Sampling details were specified more clearly 130-133

  1. 115: Was parental consent obtained for participants who were younger than 18 years old?

Information about consent was included 159-160

  1. 118: I suggest using a different term than “sportsmen” since the sample also included females.

Changed according to suggestion

  1. 140-180: Please make sure to provide response format, example items, and psychometrics for all instruments.

Examples of answers and response format was included. All test have Cronbach alpha was provided.

  1. 157-158: The author states that “a list of twelve motivations was created and their importance was assessed by the subjects” (ll. 157-158). How does this differ from the original instrument and why were modifications done?

Rationale for modification was included and changed statements were presented in the text.

RESULTS

Specific comments:

Tables 1 and 2: I think it is necessary to provide descriptive statistics for all assessed variables.

Descriptive statistics was provided in additional tables

  1. 259-260 and 274-275: Based on gender?

Information was added to the text

DISCUSSION

The Discussion includes various interpretations that are not supported by empirical evidence. I would like to see the author provide such links when offering explanations for the current findings.

We agree with this remark of reviewer and tried to provide links to findings, but sometimes we didn’t find scientific evidence for discussion. In these parts we have changed the style of discussion for “speculating” about explanation.

Specific comments:

  1. 280-307: So how do you interpret the differences between men and women in your sample? I do not believe that ll. 282-284 is sufficient in explaining the current findings.

Unfortunately we didn’t find additional explanation for those differences in empirical evidence. We have changed assumption to speculation.

  1. 295-301: Please clarify as these statements seem to contradict each other. Then, please also revise the rest of this paragraph (ll. 302-307) for clarity.

Paragraph about tendency for addiction was re-written and we’ve added limitation based on questionnaire specifics

  1. 308: What constitutes a “high result?”

High result is considered 8, 9 or 10 in sten norms.

  1. 308-399: Please split this extremely long paragraph.

Done according to the suggestion of reviewer

  1. 315: Surprising in what way?

The sentence was reformulated.

  1. 321: “state of affairs”?

changed to previous findings

  1. 321-323: Would this not suggest that personality does not determine participation then?

Added the sentence to clarify this suggestion

  1. 323-334: How do you interpret this relationship? Please go beyond the descriptive comparison to previous studies.

Paragraph about tendency for addiction was re-written and we’ve added limitation based on questionnaire specifics

  1. 347-352: Please provide supporting evidence.

Provided article we’ve based our assumption on.

  1. 356-357: Please explain further and provide supporting evidence

we couldn’t find evidence supporting this sentence, so this speculation was deleted from the text.

  1. 361-362: What constitutes a “responsible and professional approach towards sport?”

explanation was added 424

  1. 380-384: Please interpret further and provide supporting evidence.

Changed according to suggestion 444

  1. 348-385: Please link with current findings.

Some sentences were rewritten in order to point out the links in clearer way.

  1. 386-389: Please clarify.

Unclear sentence was deleted from the text

  1. 391-392: Please provide reference.

we couldn’t find evidence supporting this sentence, so this speculation was deleted from the text.

Round 2

Reviewer 1 Report

** Please refer reviewer's desriptions to overall sentences

* Line: Revision required descriptions 

-------------------------------------------------------------------------------

** It seems better to use the same form of either "sport" or "sports" from the beginning to the end of the paper.   

* 44: Prabably "less dangerous than ES and practiced" rather than "least dangerous then ES and are practiced"

* 45: "The type of these disciplines" or "These types of the discipline" rather than "These type of disciplines" 

* 46: " while" rather than ". While"

* 48-49: "one of the main goals" rather than "one of main goals"

* 49: "those types of sports" or the typr of those sports" rather than "those type of sport" 

* 50: "This determination" rather than "This distinction"

* 50: "better understanding"  rather than "a better understanding"

** 50: the clause "understaning of complexity of the phenomenon" seems confused. Does it mean that "better understanding of the contexts of the distinction"?  

** 51: "the diversity in personality determinants and motivations" --> "diversity of the personalilty determinants and the motivation types (forms or kinds)"

* 55: "Already by the 1970's people were using" --> "Already by the 1970's,"

* 63-66: "Thus, sports in which the performance lasts a few seconds can be considered a strong source of stimulation, while sports in which the action takes much longer, are more often associated with the ability to endure discomfort and sometimes avoid stimulation" --> "Thus, sports in which the performance lasts a few seconds can be considered a source of strong stimulation while sports in which the action takes much longer are more often associated with the ability to endure discomfort and sometimes eveen avoid stimulation"

** 63-66: The revised sentence is proposed right above, but this sentence is still hardly understood.    

** 85: The clause ", according to which" needs to be revised.

* 86: "chooses goals, palns their" --> "chooses goals and plans their" 

Author Response

Cover letter after review.

Once again we would like to thank the reviewers for in depth and helpful comments about our paper. We have tried to answer them with regards to our best knowledge and experience. We hope that our answers satisfy the reviewers. Since one of the reviewers recommended that our manuscript should undergo extensive English revisions, we are willing to do so, once our revisions are accepted by the editors and the reviewers.

Now we would like to answer point by point the details of the revision.

Answer to the Reviewer 1.

Thank you for your suggestions, we accept all the remarks and changed them in the text. Moreover we are willing to send our paper to extensive English revisions by MDPI, after acceptance of our revisions by editors and second reviewer.

** It seems better to use the same form of either "sport" or "sports" from the beginning to the end of the paper.   

* 44: Prabably "less dangerous than ES and practiced" rather than "least dangerous then ES and are practiced"

* 45: "The type of these disciplines" or "These types of the discipline" rather than "These type of disciplines" 

* 46: " while" rather than ". While"

* 48-49: "one of the main goals" rather than "one of main goals"

* 49: "those types of sports" or the typr of those sports" rather than "those type of sport" 

* 50: "This determination" rather than "This distinction"

* 50: "better understanding"  rather than "a better understanding"

** 50: the clause "understaning of complexity of the phenomenon" seems confused. Does it mean that "better understanding of the contexts of the distinction"?  

** 51: "the diversity in personality determinants and motivations" --> "diversity of the personalilty determinants and the motivation types (forms or kinds)"

* 55: "Already by the 1970's people were using" --> "Already by the 1970's,"

* 63-66: "Thus, sports in which the performance lasts a few seconds can be considered a strong source of stimulation, while sports in which the action takes much longer, are more often associated with the ability to endure discomfort and sometimes avoid stimulation" --> "Thus, sports in which the performance lasts a few seconds can be considered a source of strong stimulation while sports in which the action takes much longer are more often associated with the ability to endure discomfort and sometimes eveen avoid stimulation"

** 63-66: The revised sentence is proposed right above, but this sentence is still hardly understood.    

** 85: The clause ", according to which" needs to be revised.

* 86: "chooses goals, palns their" --> "chooses goals and plans their" 

Reviewer 2 Report

INTRODUCTION

I appreciate the revisions that were made to the manuscript. Unfortunately, my most significant concern was not addressed. I previously stated that “While the Introduction provides helpful background information for the current study, it currently fails to establish a strong rationale for the research. Specifically, what are the gaps/limitations in previous literature that are being addressed. In other words, why was the current study needed? The information in lines 92-94 is not sufficient in establishing this rationale.” I was looking for a major revision here to fully develop this rationale so the three-line addition the authors mention in their point-by-point response is not sufficient.

Specific comments:

ll. 63-66: Please provide supporting reference.

ll. 84-87: If this is the theoretical framework of the current study, a more in-depth explanation is needed. Overall, the conceptual information in the Introduction is lacking.

METHOD

Specific comments:

ll. 210-212: Please provide more detail.

ll. 127-129: Please provide rationale for restricting recruitment to these clubs. Information regarding sampling procedures is still missing.

ll. 181-182: Please provide more detail regarding these modifications as well as a stronger rationale for altering the original instrument.

DISCUSSION

Specific comments:

ll. 330-332: The authors state in their point-by-point response that they consider a “high result” scores of 8-10. What is this classification based on?

ll. 343-344: This interpretation appears incomplete and needs to be expanded on.

Author Response

Cover letter after review.

Once again we would like to thank the reviewers for in depth and helpful comments about our paper. We have tried to answer them with regards to our best knowledge and experience. We hope that our answers satisfy the reviewers. Since one of the reviewers recommended that our manuscript should undergo extensive English revisions, we are willing to do so, once our revisions are accepted by the editors and the reviewers.

Now we would like to answer point by point the details of the revision.

Answer to the Reviewer 2

Once again, we wanted to express our gratitude for the accurate and relevant comments regarding our article.

I appreciate the revisions that were made to the manuscript. Unfortunately, my most significant concern was not addressed. I previously stated that “While the Introduction provides helpful background information for the current study, it currently fails to establish a strong rationale for the research. Specifically, what are the gaps/limitations in previous literature that are being addressed. In other words, why was the current study needed? The information in lines 92-94 is not sufficient in establishing this rationale.” I was looking for a major revision here to fully develop this rationale so the three-line addition the authors mention in their point-by-point response is not sufficient.

We tried to enrich the argumentation and give broader rationales of the study. We tried to highlight the gaps in the research so far:

- temperament studies based mainly on SSS (ceiling effect)

- research on motives mainly in the qualitative paradigm (few quantitative studies)

- lack of a holistic model combining personality, motivation and a sense of satisfaction with life

- the lack of distinction between HRS and ES, which may differ in both danger and social attractiveness.

We hope that this rationale satisfies the reviewer.

Specific comments:

  1. 63-66: Please provide supporting reference.

Supporting evidence provided (line 66)

  1. 84-87: If this is the theoretical framework of the current study, a more in-depth explanation is needed. Overall, the conceptual information in the Introduction is lacking.

More in-depth explanation is provided (lines 96-105)

METHOD

Specific comments:

  1. 210-212: Please provide more detail.

More details are provided (246-253)

  1. 127-129: Please provide rationale for restricting recruitment to these clubs. Information regarding sampling procedures is still missing.

Rationale included, also purposeful sampling procedures added to limitations of the study 148-150, 487-488

  1. 181-182: Please provide more detail regarding these modifications as well as a stronger rationale for altering the original instrument.

Additional rationale included 207-215

DISCUSSION

Specific comments:

  1. 330-332: The authors state in their point-by-point response that they consider a “high result” scores of 8-10. What is this classification based on?

Explanation added to instruments section 196-197

  1. 343-344: This interpretation appears incomplete and needs to be expanded on.

The interpretation was added (381-385)

Round 3

Reviewer 2 Report

INTRODUCTION

The rationale that is being developed in the Introduction still needs to be stronger. I continue to look for a major revision and the authors continue to add single sentences which are not sufficient. The gaps the authors list in their point-by-point response are relevant, but they need to be incorporated, explained, and supported further in the Introduction.

METHOD

Specific comments:

ll. 148-150: Not sure this justification makes sense to me. Would an online survey not be more viable for sensitive data due to its anonymous nature? And, what financial resources would have been needed for that type of data collection?

l. 207: Please provide references and respective findings that support this decision.

Author Response

Answer to reviewer 3

INTRODUCTION

The rationale that is being developed in the Introduction still needs to be stronger. I continue to look for a major revision and the authors continue to add single sentences which are not sufficient. The gaps the authors list in their point-by-point response are relevant, but they need to be incorporated, explained, and supported further in the Introduction.

 Thank you for pointing out things that can improve our paper. I must admit I’ve tried to build clear and scientific rationale and build the introduction around it (We’ve covered – the definition of HRS and ES, current research of personality, motives and satisfaction with life). I’ve added some sentences that make the introduction clearer. I can imagine that it’s unfortunately not, what the Reviewers expects of this paper. Since I have no idea, how to further improve the introduction I kindly ask for direct suggestions how to improve or to consider if this state of introduction is sufficient for the Journal.

METHOD

Specific comments:

  1. 148-150: Not sure this justification makes sense to me. Would an online survey not be more viable for sensitive data due to its anonymous nature? And, what financial resources would have been needed for that type of data collection?

The questionnaire used in the study was acquired from the laboratory in paper-pencil style, while online version required an expensive platform to used, which was out of the budget range of our study. Apart from that, after completing the questionnaires and counting the results and had the opportunity to discuss the results of the personality test live in an individual conversation with the psychologist conducting the study.

  1. 207: Please provide references and respective findings that support this decision.

Since the author of the research tool is also a co-author of the presented study, we decided to change the selected points after conducting the pilot study. The pilot study has not been published, so we are unable to provide a reference.

I hope these arguments are sufficient for the reviewer.

Best regards,
